# Targeting plasmid-encoded proteins that contain immunoglobulin-like domains to combat antimicrobial resistance

Alejandro Prieto[1†], Luïsa Miró[2,3†], Yago Margolles[4], Manuel Bernabeu[1], David Salguero[1], Susana Merino[1], Joan Tomas[1], Juan Alberto Corbera[5], Anna Perez-Bosque[2,3], Mario Huttener[1], Luis Ángel Fernández[4], Antonio Juarez[1,6*]

[1]Department of Genetics, Microbiology and Statistics, University of Barcelona, Barcelona, Spain; [2]Department of Biochemistry and Physiology, Universitat de Barcelona, Barcelona, Spain; [3]Institut de Nutrició i Seguretat Alimentària, Universitat de Barcelona, Barcelona, Spain; [4]Department of Microbial Biotechnology, Centro Nacional de Biotecnología, Consejo Superior de Investigaciones Científicas (CNB-CSIC), Madrid, Spain; [5]Instituto Universitario de Investigaciones Biomédicas y Sanitarias (IUIBS), Facultad de Veterinaria, Universidad de Las Palmas de Gran Canaria (ULPGC), Campus Universitario de Arucas, Las Palmas, Spain; [6]Institute for Bioengineering of Catalonia, The Barcelona Institute of Science and Technology, Barcelona, Spain

*For correspondence:
ajuarez@ub.edu

†These authors contributed equally to this work

Competing interest: The authors declare that no competing interests exist.

**Abstract** Antimicrobial resistance (AMR) poses a significant threat to human health. Although vaccines have been developed to combat AMR, it has proven challenging to associate specific vaccine antigens with AMR. Bacterial plasmids play a crucial role in the transmission of AMR. Our recent research has identified a group of bacterial plasmids (specifically, IncHI plasmids) that encode large molecular mass proteins containing bacterial immunoglobulin-like domains. These proteins are found on the external surface of the bacterial cells, such as in the flagella or conjugative pili. In this study, we show that these proteins are antigenic and can protect mice from infection caused by an AMR *Salmonella* strain harboring one of these plasmids. Furthermore, we successfully generated nanobodies targeting these proteins, that were shown to interfere with the conjugative transfer of IncHI plasmids. Considering that these proteins are also encoded in other groups of plasmids, such as IncA/C and IncP2, targeting them could be a valuable strategy in combating AMR infections caused by bacteria harboring different groups of AMR plasmids. Since the selected antigens are directly linked to AMR itself, the protective effect extends beyond specific microorganisms to include all those carrying the corresponding resistance plasmids.

## eLife assessment

This **important** and novel study addresses the challenge of antimicrobial resistance by targeting plasmid proteins that interfere with plasmid transfer as a strategy to limit the spread of antibiotic-resistance genes. The evidence presented and the integration of two approaches to tackle antimicrobial resistance is **convincing**. This work will interest those working on plasmid transfer and antimicrobial resistance.

## Introduction

Despite the availability of antibiotics, bacterial infectious diseases are the second leading cause of death worldwide (*Morens et al., 2004*; *Murray et al., 2022*). The gradual increase in the resistance rates of several important bacterial pathogens represents a serious threat to public health (*Meyer et al., 2010*; *Rossolini et al., 2007*; *Spellberg et al., 2008*). In 2019, up to 495 million deaths could be associated with AMR (*Murray et al., 2022*).

Plasmids can confer resistance to the major classes of antimicrobials (*Carattoli, 2009*). Their transmission by horizontal gene transfer (HGT) is largely underlying the dissemination of AMR genes, especially in Gram-negative bacteria (*Carattoli, 2013*; *Thomas and Nielsen, 2005*; *Wang et al., 2015*), but also in Gram-positive bacteria (*Vrancianu et al., 2020*). Among the various methods utilized for classifying plasmids, grouping them into incompatibility groups (Inc) is a well-established and extensively employed approach (*Novick and Richmond, 1965*).

The incompatibility group HI includes plasmids that are widespread in the *Enterobacteriaceae*. These plasmids frequently include genetic elements that encode multiple AMR determinants (*Phan and Wain, 2008*). IncHI-encoded AMR can be present in enterobacteria such as *Salmonella* (*Phan and Wain, 2008*), *Escherichia coli* (*Forde et al., 2018*), *Klebsiella pneumoniae* (*Villa et al., 2012*), and *Citrobacter freundii* (*Dolejska et al., 2013*). Several *Salmonella* isolates harbor IncHI plasmids, with those of the IncHI2 subgroup prevailing in antibiotic-resistant *Salmonella* isolates. In *S. enterica* serotype Typhi, over 40% of isolates are found to carry an IncHI plasmid (*Holt et al., 2011*). Recent reports have shown a novel role of IncHI plasmids in AMR spread. Colistin is a last-resort antibiotic for the treatment of severe infections (*Lim et al., 2010*). Plasmid-mediated resistance, conferred by the mobilized colistin resistance gene (*mcr-1*), has emerged recently. Approximately 20% of global isolates carrying the *mcr-1* gene are associated with an IncHI2 plasmid, and this percentage escalates to 41% when considering exclusively European isolates (*Matamoros et al., 2017*). Of special concern is the presence of the *mcr-1* resistance determinant in enterobacterial isolates carrying carbapenem resistance genes, such as *blaNDM* and *blaKPC*. The combination of these AMR determinants can seriously compromise the treatment of infections caused by virulent strains harboring these plasmids (*Wang et al., 2017*; *Zheng et al., 2016*). An example of this is the recent report of an AMR clone of the highly virulent *E. coli* ST95 lineage (*Forde et al., 2018*). *E. coli* ST95 clones cause neonatal meningitis and sepsis. They are usually sensitive to several antibiotics. This AMR clone harbors an IncHI2 plasmid that carries, among other factors, genes encoding determinants of resistance to colistin and multiple other antibiotics (including the extended-spectrum beta-lactamase $bla_{CTX-M-1}$ gene cluster). The spread of such an AMR ST95 clone could pose a threat to human health worldwide (*Forde et al., 2018*).

The R27 plasmid is the best-studied IncHI1 plasmid. It harbors the Tn10 transposon, which confers resistance to tetracycline. The R27 replicative and conjugative machineries have been studied in detail (*Lawley et al., 2003*; *Lawley et al., 2002*) and its complete nucleotide sequence is known (*Sherburne et al., 2000*). A newly reported feature of IncHI plasmids is that they code for large molecular mass proteins that contain bacterial Ig-like (Big) domains (*Hüttener et al., 2019*). Bacterial proteins containing Big domains play different roles in the cell. Some fimbrial subunits, adhesins, membrane transporters, and enzymes contain Big domains (as reviewed in *Bodelón et al., 2013*). The IncHI1 plasmid R27 encodes the *rsp* and *rsp*2 genes which code respectively for the 155.4 kDa RSP and 86.75 kDa RSP2 proteins. These proteins interact and target both the flagella and the conjugative pili to favor conjugation (*Hüttener et al., 2022*). Big proteins are also encoded at least by two other groups of plasmids (IncA/C and IncP2) (*Hüttener et al., 2022*). Due to their extracellular location, these plasmid-encoded Big proteins could be considered as valuable antigens to combat antimicrobial resistance.

Functional heavy-chain-only antibodies (HCAbs) were discovered nearly 30 years ago in the serum of camelids (*Hamers-Casterman et al., 1993*). HCAbs lack both the light chains and the first constant CH1 domain within the heavy chain. The variable domain of HCAbs, also termed VHH for <u>V</u>H of <u>H</u>CAb, represents the smallest antigen-binding unit that occurs in nature (*Muyldermans, 2013*). The VHH domains can be expressed in recombinant form as highly stable single-domain Ab fragments of ~14 kDa, also termed Nanobodies (Nbs), which exhibit high affinity and specificity to its cognate antigens and unique recognition of cryptic epitopes not recognized by conventional Abs (*Muyldermans, 2013*). Nbs can be easily produced in bacteria and other microbial cultures, and used in multiple diagnostic and therapeutic applications, including bacterial infections (*Muyldermans, 2021*).

In this study, our objective was to investigate the potential of the recently discovered Big proteins encoded by IncHI plasmids as targets for combating antibiotic resistance. Our research focused on two key aspects. First, we examined the efficacy of the RSP protein as a vaccine antigen in providing protection against infections caused by resistant bacteria carrying an IncHI plasmid. The results obtained confirmed the efficacy of the RSP protein as a vaccine antigen. Furthermore, we also generated RSP-specific Nbs and showed that they can interfere with the conjugative transfer of the R27 plasmid. Hence, the use of antigens that are directly linked to AMR determinants can be effective in combating infections caused by AMR bacteria.

## Results

### Ampicillin protects mice from infection with the *S.* Typhimurium SL1344 (WT) strain, but not from infection with the *S.* Typhimurium SL1344 (pHCM1) strain

Considering the prevalence of IncHI plasmids in *Salmonella,* and the use of *S.* Typhimurium infection in mice as a model for *Salmonella* infection, we employed a mouse infection model to validate the RSP protein as a vaccine antigen against an antibiotic-resistant *Salmonella* strain. For antibiotic treatment, we specifically selected ampicillin (Amp) due to its established efficacy in reducing the symptoms of

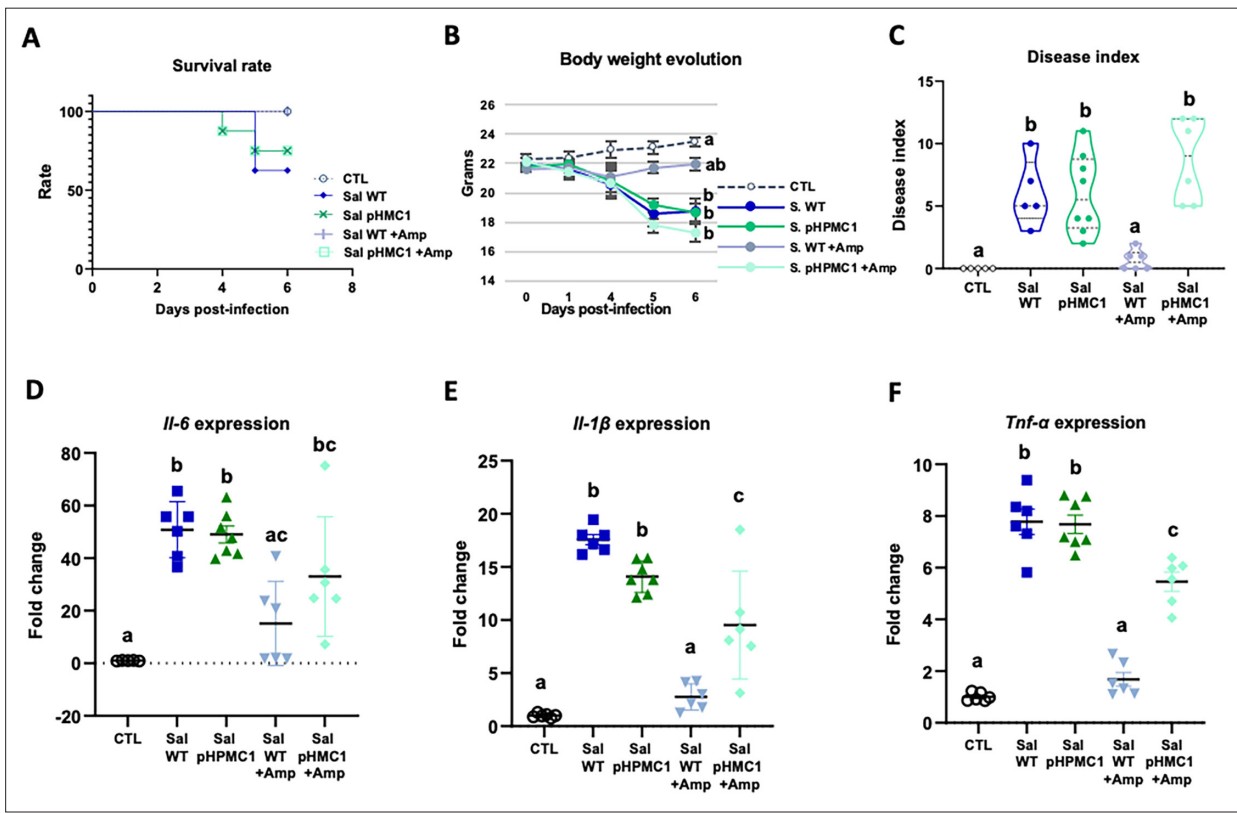

**Figure 1.** Infection of mice with the *S.* Typhimurium SL1344 and *S.* Typhimurium (pHCM1) strains. Survival rate (**A**), body weight evolution (**B**), disease index (**C**), and proinflammatory cytokine expression in spleen, *Il-6* (**D**), *Il-1β* (**E**), and *Tnf-α* (**F**). Groups treated with ampicillin (Amp) are indicated with + Amp. In panel (**A**), the survival rate of control animals and those infected with the *S.* Typhimurium SL1344 (WT) strain and treated with Amp, are 100%. Animals infected with the *S.* Typhimurium (pHCM1) strain have the same survival rate independently of the treatment with Amp. Results are expressed as mean ± SEM (n=6–8 animals). Means without a common letter differ, p<0.05. The survival rates were compared by Log-rank (Mantel-Cox) test and expressed as the percentage of survival. Body weight evolution was analyzed by means of repeated measures ANOVA. Disease indices are expressed as median values and quartiles, and were compared by the Kruskal-Wallis test. The expression of cytokines were analyzed by using one-way analysis of variance (ANOVA) followed by Fisher's least significant difference (LSD) post hoc test.

The online version of this article includes the following source data for figure 1:

**Source data 1.** The Excel file containing numerical source data of the results is shown in **Figure 1**.

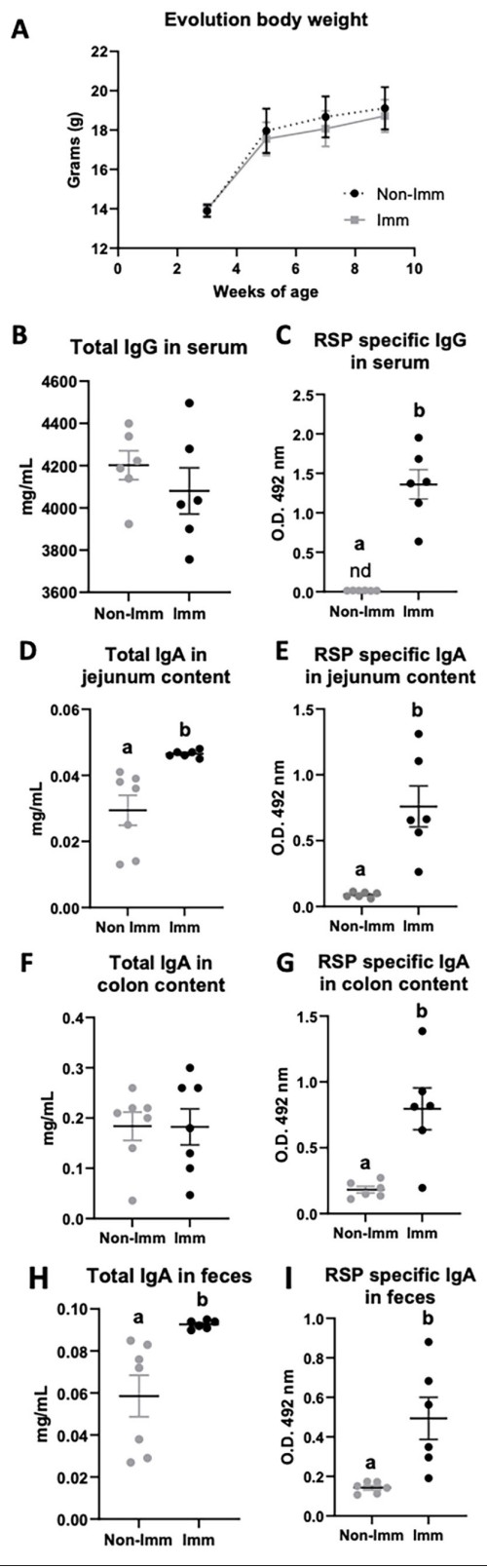

**Figure 2.** Effect of immunization with the RSP protein on body weight evolution (**A**) and immunoglobulin concentration (**B–I**). Grey symbols represent non-immunized (**No-Imm**) mice; black symbols represent

*Figure 2 continued on next page*

immunized (**Imm**) mice. Results are expressed as mean ± SEM (n=6–8 animals). IgA, immunoglobulin A; IgG, immunoglobulin G; nd, non-detected. Means without a common letter differ, p<0.05. Data were analyzed with Student's t-test.

The online version of this article includes the following source data for figure 2:

**Source data 1.** The Excel file containing numerical source data of the results is shown in **Figure 2**.

*Salmonella* infection in mice (**Butler et al., 1981**; **Rossi et al., 2017**). For our study, we chose the IncHI plasmid pHCM1, which harbors an Amp resistance gene.

We first established that both *Salmonella* strains used (WT and pHCM1) were capable of infecting mice, and that only the plasmid-bearing strain was antibiotic resistant. Infection with the *Salmonella* strain SL1344 (WT) reduced the survival rate of the mice by about 40% (**Figure 1A**, **Figure 1—source data 1**), while the addition of Amp increased the survival rate of these animals. On the other hand, Amp treatment did not change the survival of mice infected with strain SL1344 (pHCM1).

In antibiotic-naive mice infected with either SL1344 (WT) or SL1344 (pHCM1) strains, gradual weight loss, and clinical signs of disease were observed (**Figure 1B and C**, **Figure 1—source data 1**). However, Amp treatment only ameliorated these symptoms in mice infected with the SL1344 (WT) strain, showing no effect on those infected with the SL1344 (pHCM1) strain. Additionally, infection with both *Salmonella* strains led to an upregulation of proinflammatory cytokines in the spleen (**Figure 1D**, **E and F**, **Figure 1—source data 1**). Amp treatment in mice infected with the SL1344 (WT) strain prevented this increase in cytokine expression.

## Immunization of mice with the RSP protein

We then purified the RSP protein and employed it as a vaccine antigen to evaluate its efficacy in protecting mice against infection caused by the *S.* Typhimurium SL1344 strain carrying the pHCM1 plasmid. To prevent the recurrence of infection by SL1344 clones that spontaneously cured the pHCM1 plasmid in the absence of Amp, we maintained the antibiotic treatment in the immunized animals.

The immunization process did not have a significant impact on the body weight progression of the mice (**Figure 2A**, **Figure 2—source data 1**).

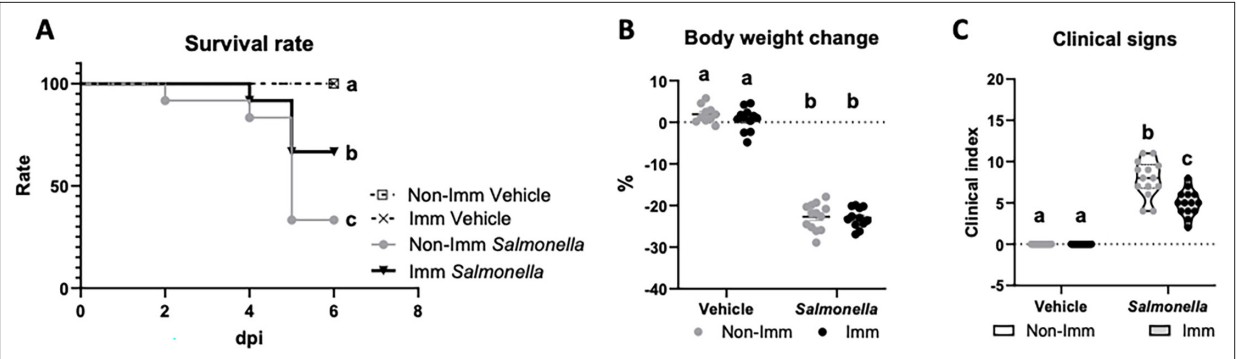

**Figure 3.** Survival rate (**A**), body weight change (**B**), and clinical signs (**C**) after challenge with the SL1344 (pHCM1) strain. Grey symbols represent non-immunized (**No-Imm**) mice; black symbols represent immunized (**Imm**) mice. In panel B, results are expressed as mean ± SEM (n=12–14 animals). Means without a common letter differ, p<0.05. The survival rates were compared by Log-rank (Mantel-Cox) test and expressed as the percentage of survival. Disease indices are expressed as median values and quartiles, and were compared by the Kruskal-Wallis test.

The online version of this article includes the following source data for figure 3:

**Source data 1.** The Excel file containing numerical source data of the results is shown in *Figure 3*.

---

Although the overall concentration of total serum IgG remained unchanged following immunization (*Figure 2B*, *Figure 2—source data 1*), there was a substantial increase in the levels of specific IgG antibodies against the RSP protein (*Figure 2C*, *Figure 2—source data 1*). In terms of intestinal IgA, immunization resulted in an elevated concentration of total IgA in both the jejunum content and feces (*Figure 2D and H*, respectively, *Figure 2—source data 1*), while not affecting its presence in the colon (*Figure 2F*, *Figure 2—source data 1*). Moreover, immunization led to enhanced production of specific IgA antibodies against the RSP protein in all three tissue types (intestinal contents *Figure 2E*; colon contents *Figure 2G*; and feces *Figure 2I*, *Figure 2—source data 1*).

### Effects of immunization of mice on the infection caused by the *S. Typhimurium* SL1344 (pHCM1) strain

#### Survival, body weight evolution, and clinical signs

Infection with the SL1344 (pHCM1) strain resulted in a decrease in the survival rate of mice and an approximate 20% reduction in their body weight after 6 days of infection. Additionally, it led to an increase in the manifestation of clinical signs associated with infection (*Figure 3*, *Figure 3—source data 1*). However, immunization significantly improved the survival rate of mice and reduced the incidence of clinical signs in infected animals.

#### Intestinal effects of immunization with the RSP protein and challenge with the SL1344 (pHCM1) strain

Immunization with the RSP protein increased the secretion of total IgA into the lumen of the jejunum and colon (*Figure 4A and B*, *Figure 4—source data 1*), while reducing its concentration in feces (*Figure 4C*, *Figure 4—source data 1*). Furthermore, immunization with the RSP protein led to an elevated presence of specific IgA antibodies against the RSP protein in all three samples (*Figure 4D–F*, *Figure 4—source data 1*), with even higher levels observed in infected animals in the case of fecal content. Regarding the expression of proinflammatory cytokines, infection with the SL1344 (pHCM1) strain resulted in increased expression of all three cytokines analyzed (*Figure 4G–I*, *Figure 4—source data 1*) both in immunized and non-immunized animals. In the case of Tnf-$\alpha$, immunization also increased its expression.

#### Systemic effects of immunization with the RSP protein and challenge with the SL1344 (pHCM1) strain

The immunization protocol also resulted in an increase in the concentration of total IgG at the systemic level (*Figure 5A*, *Figure 5—source data 1*), while infection with the SL1344 (pHCM1) strain reduced it. Additionally, immunization also led to an elevated concentration of specific anti-RSP IgG in the

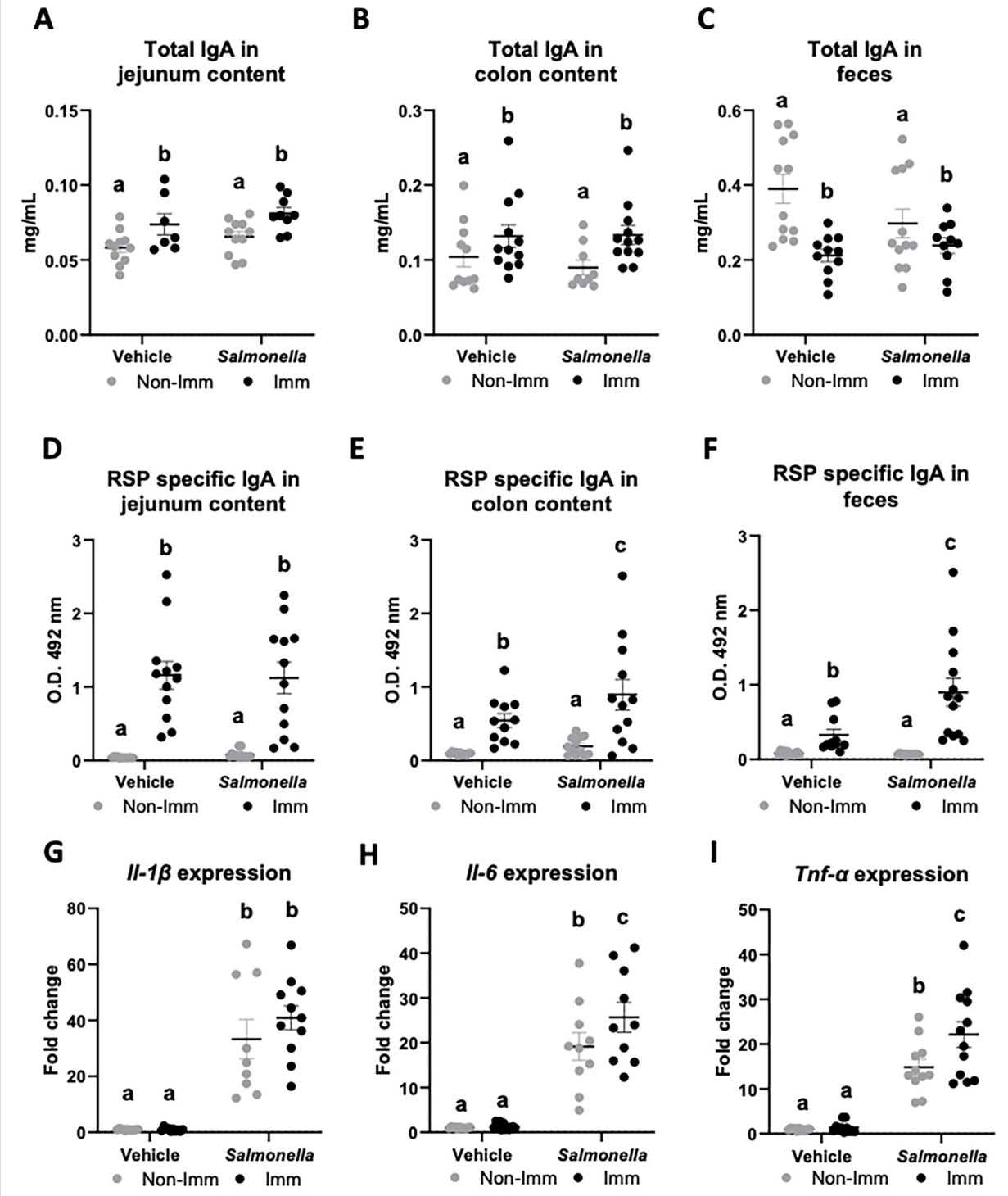

**Figure 4.** Immunoglobulin concentration and cytokine expression in the gastrointestinal tract after RSP immunization and challenge with *Salmonella*. Total IgA concentration in jejunum content (**A**), in colon content (**B**), and in feces (**C**). RSP-specific IgA titers in jejunum content (**D**), in colon content (**E**), and in feces (**F**). *Il-1β* (**G**), *Il-6* (**H**), and *Tnf-α* (**I**) expression in colon mucosa. Open bars represent non-immunized (**Non-Imm**) mice; solid bars represent immunized (**Imm**) mice. Results are expressed as means ± SEMs (n=10–12 animals). Means without a common letter differ, p<0.05. IgA, immunoglobulin A; IgG, immunoglobulin G; Il, interleukin; Int, the interaction between both factors; Tnf-α, tumor necrosis factor-alpha. All data were analyzed using two-way ANOVA (Immunization and infection factors).

The online version of this article includes the following source data for figure 4:

**Source data 1.** The Excel file containing numerical source data of the results is shown in *Figure 4*.

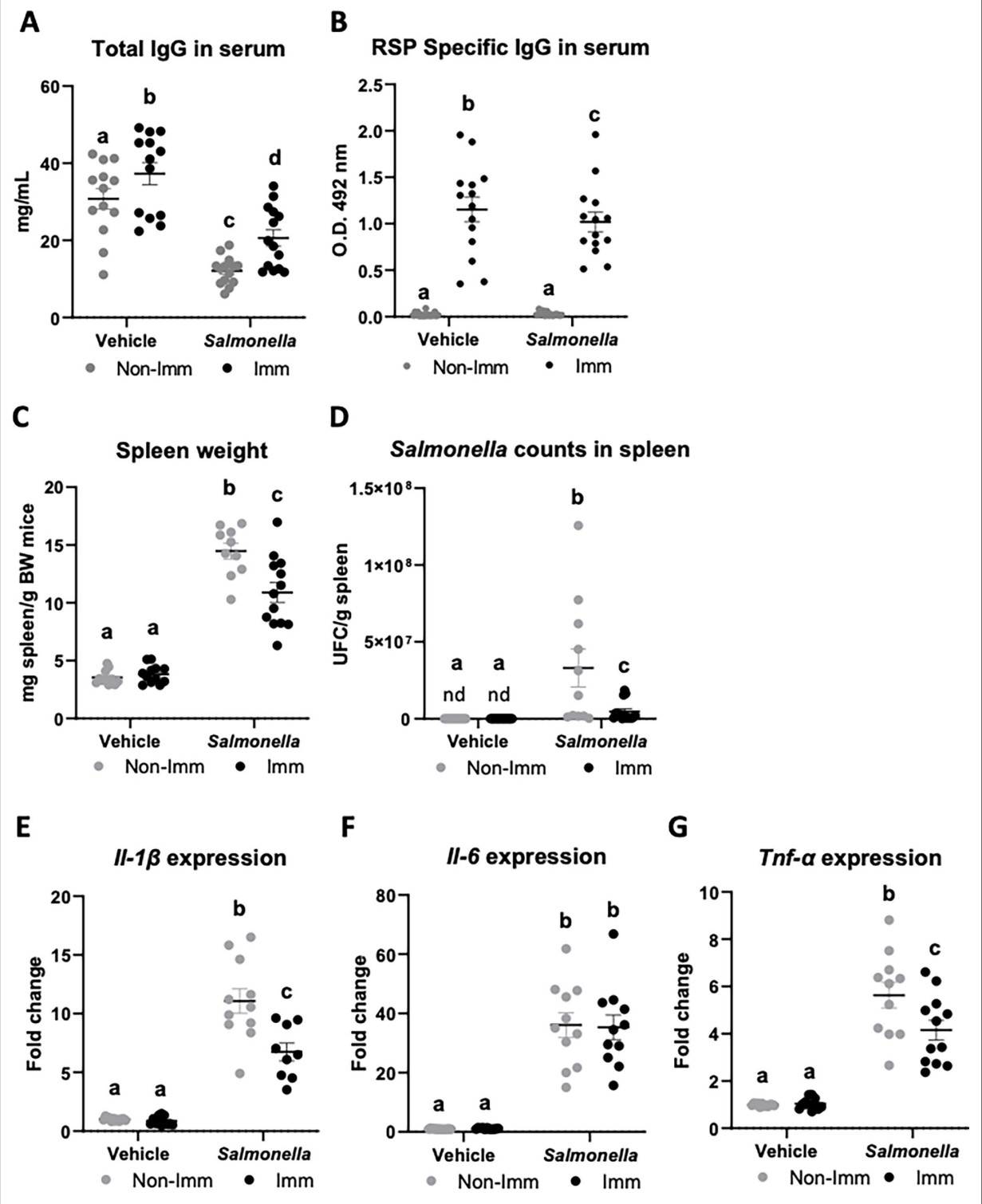

**Figure 5.** Immunoglobulin concentration, spleen weight, *Salmonella* counts in spleen tissue, and cytokine expression in spleen tissue after immunization with the RSP protein and challenge with the SL1344 (pHCM1) strain. Total IgG concentration (**A**) and RSP-specific IgG titers (**B**) in plasma. Spleen weight (**C**) and *Salmonella* counts in spleen tissue (**D**). *Il-1β* (**E**), *Il-6* (**F**) and *Tnf-α* (**G**) expression in spleen tissue. Results are expressed as means ± SEMs (n=12–14 animals). Means without a common letter differ, p<0.05. IgA, immunoglobulin A; IgG, immunoglobulin G; Il, interleukin; nd, non-detected; Tnf-α, tumor necrosis factor-alpha. All data were analyzed using two-way ANOVA (Immunization and infection factors).

The online version of this article includes the following source data for figure 5:

**Source data 1.** The Excel file containing numerical source data of the results is shown in *Figure 5*.

serum (*Figure 5B*, *Figure 5—source data 1*), which remained unaffected after infection with the SL1344 (pHCM1) strain. Following bacterial challenge, the spleen weight increased nearly threefold in non-immunized mice (*Figure 5C*, *Figure 5—source data 1*). In contrast, immunized mice exhibited a significantly lower increase in spleen weight. Furthermore, animals infected with the SL1344 (pHCM1) strain displayed *Salmonella* colonization in the spleen (*Figure 5D*, *Figure 5—source data 1*), which was significantly reduced in immunized animals. Regarding the expression of proinflammatory cytokines in the spleen, *Salmonella* infection upregulated the expression of all cytokines evaluated (*Figure 5E–G*, *Figure 5—source data 1*). Immunization partially mitigated this effect on Il-1β and Tnf-α expression.

## Selection of nanobodies with neutralizing activity against the RSP protein

Given the protective effect against *Salmonella* infection observed in mice immunized with the RSP protein, we wondered whether we could obtain specific antibodies against RSP that could directly interfere with its biological activity in the conjugative transfer of IncHI plasmids. We chose nanobodies as ideal candidates to block RSP activity given their small size, stability, and high affinity to antigens, and their therapeutic potential against infections. Interestingly, nanobodies can also be expressed on the surface of *E. coli* fused to the initial 654 amino acids of the enterohemorrhagic *E. coli* intimin protein, termed Neae (*Bodelón et al., 2009*), comprising the N-terminal signal peptide, the periplasmic LysM domain, the β-barrel for outer membrane insertion, and the extracellular Big domains D00 and D0 (*Fairman et al., 2012*). Expression of Neae-Nb fusions represents an efficient bacterial display system for their selection of Nbs binding a target antigen from VHH gene libraries employing *E. coli* cell sorting methods with the labeled antigen (*Salema et al., 2013*; *Salema and Fernández, 2017*). In addition, Neae-Nb fusions expressed in *E. coli* promote synthetic cell-to-cell adhesions mediated by the specific recognition of a surface antigen in the target cell (*Glass and Riedel-Kruse, 2018*; *Piñero-Lambea et al., 2015*).

In order to obtain Nbs that specifically bind the extracellular domain of the RSP protein, we immunized two dromedaries with the last 280 residues of the C-terminal domain of the RSP protein. After immunization, peripheral blood lymphocytes from these animals were isolated and the VHH gene segments were amplified and cloned in the display vector pNeae2 (*Robledo et al., 2022*; *Salema et al., 2013*), which encodes the Neae intimin fragment under the control of the IPTG-inducible lacI-Plac promoter region. A library of approximately $1 \times 10^8$ clones was obtained by electroporation of *E. coli* DH10BT1R strain with the products of the ligated pNeae2-VHH. Bacteria in the library were grown in liquid culture and induced with IPTG. Induced bacteria binding the C-domain of RSP were selected by incubation with the purified protein labeled with biotin, following iterative steps of magnetic cell sorting (MACS) and fluorescence-assisted cell sorting (FACS) (Materials and methods). After the final sorting step (*Figure 6A*), the selected pool of bacteria was grown on LB-agar plates. 96 colonies were randomly picked from these plates, grown in liquid, and induced with IPTG to evaluate the Nb display levels on the bacterial surface and their binding capability to the biotinylated C-domain of the RSP protein by flow cytometry (*Figure 6B*). The VHH sequences from clones with specific binding to purified RSP were determined, which allowed us to identify seven different Nbs (*Figure 6—figure supplement 1*).

## Selection of Nb clones displayed on *E. coli* that agglutinate with the *S.* Typhimurium SL1344 (R27) strain

For selection of Nbs binding native RSP expressed on *Salmonella* cells, we took advantage of the specific adhesion observed between *E. coli* cells with displayed Nbs and bacteria having the target antigen accessible on their surface, which leads to the agglutination of mixed bacterial cultures (*Glass and Riedel-Kruse, 2018*; *Robledo et al., 2022*).

We initially evaluated the agglutination capability of the seven different clones obtained in the enrichment protocol of the immune library, from strain *E. coli* DH10BT1R, each carrying a pNeae2-VHH derivative expressing RSP-specific Nbs, against the *S.* Typhimurium SL1344 (R27) strain. To achieve this, we cultured the *Salmonella* strain in LB medium to an OD$_{600 \text{ nm}}$ of 2. Following centrifugation, cells were resuspended in PBS and mixed with an equal volume of a culture of the different *E. coli* DH10BT1R clones displaying the different sets of RSP-specific Nbs. Only one of the clones tested (Nb

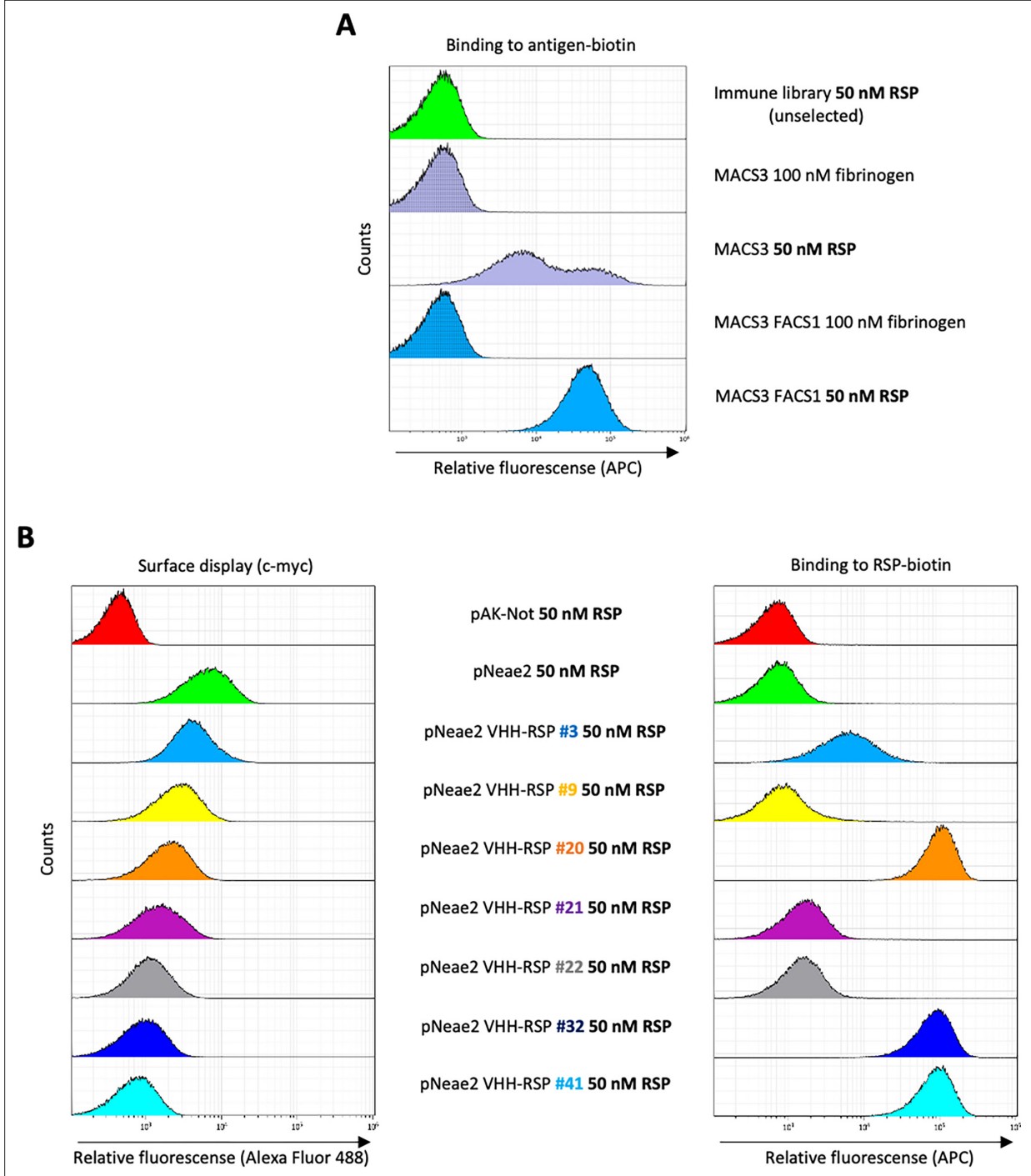

**Figure 6.** Selection of RSP-binding Nbs from immune libraries displayed on the surface of *E. coli*. (**A**) Flow cytometry analysis of the enrichment of bacterial population displaying Nb libraries generated after the immunization of dromedaries with the C-terminal domain of the RSP protein. Three rounds of selection with the RSP protein by magnetic cell sorting (MACS) and one round of fluorescence-activated cell sorting (FACS) were performed, in which bacteria were incubated with 50 nM of biotin-labeled RSP (or 100 nM of fibrinogen used as a specificity control) and stained with Streptavidin-APC. (**B**) Flow cytometry of the individual bacterial clones selected from the immune library. The bacterial surface display of the corresponding nanobody was detected using an anti-c-myc monoclonal antibody. The binding of biotin-labeled antigens to Nbs displayed on bacteria was performed with the incubation of bacterial cells with 50 nM of biotin-labeled RSP.

The online version of this article includes the following figure supplement(s) for figure 6:

**Figure supplement 1.** Amino acid sequence alignment of the selected VHHs after the magnetic cell sorting (MACS) and fluorescence-assisted cell sorting (FACS) enrichment protocol of the bacterial library generated after the immunization of two dromedaries with the RSP protein.

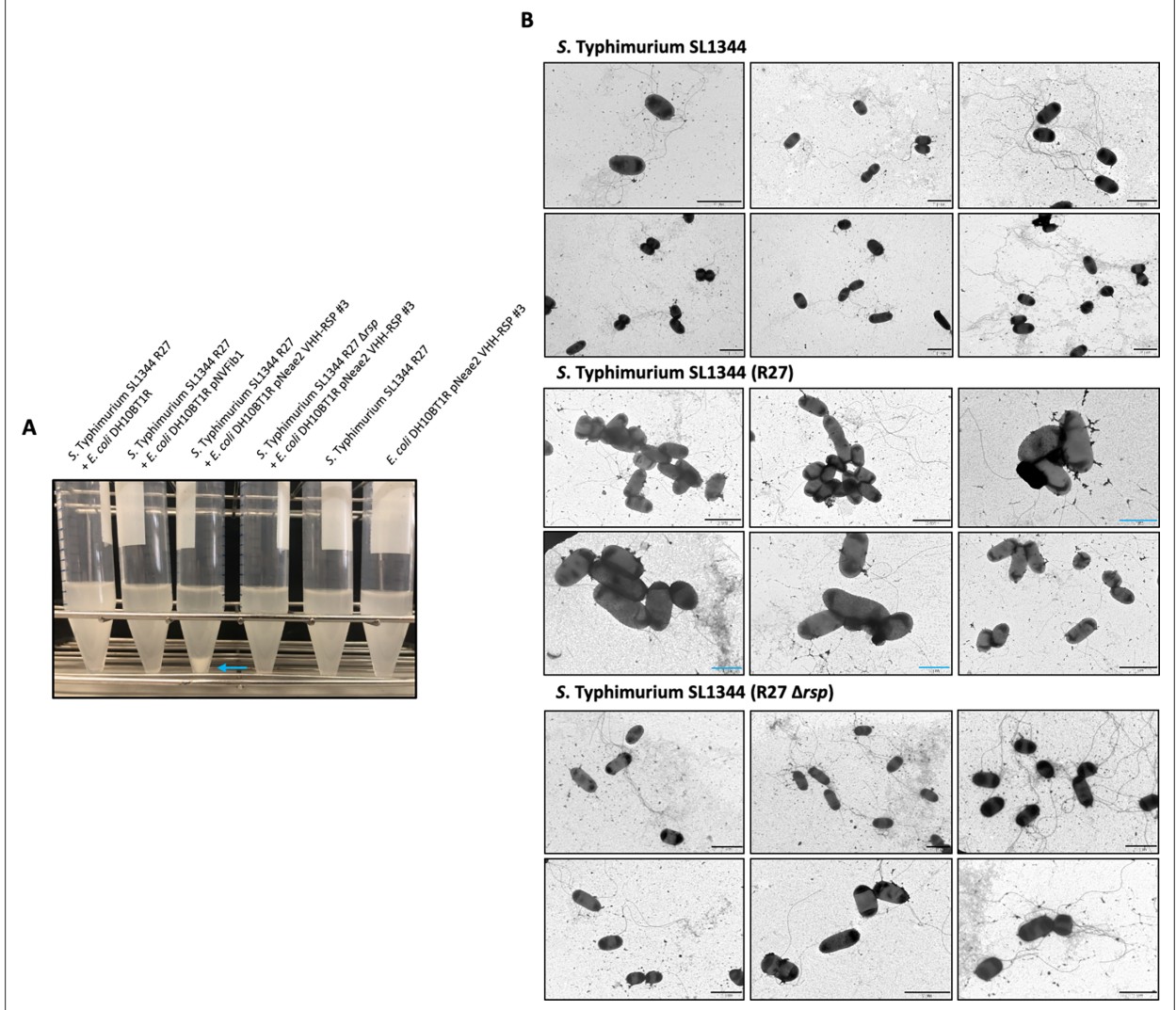

**Figure 7.** Analysis of the interaction of the VHH-RSP#3 isolated in the immune library with the native RSP protein at a macro and microscopic level. (**A**) Assay of the ability of *E. coli* DH10BT1R cells expressing nanobodies on its surface to agglutinate cells of the *S*. Typhimurium SL1344 (R27) strain. Blue arrow marks the aggregation of bacteria after the interaction of *E. coli* cells producing nanobodies directed against the RSP protein (VHH-RSP #3) and *Salmonella* cells harboring the R27 plasmid. (**B**) Transmission electron microscopy imaging mixtures of *E. coli* DH10BT1R cells expressing nanobodies against the RSP protein on its surface and cells of the *S*. Typhimurium SL1344 strain, harboring or not the R27 plasmid. The studies were performed by labeling the nanobodies-producing bacteria (*E. coli* DH10BT1R) with a mouse antibody anti c-myc-tag and goat anti-mouse IgG conjugated to 12 nm gold particles. Blue bars represent 1 µm while black bars represent 2 µm.

The online version of this article includes the following source data for figure 7:

**Source data 1.** Full-length images of transmission electron microscopy pictures shown in *Figure 7B*.

3) showed the ability of agglutinating the *S*. Typhimurium SL1344 (R27) strain (*Figure 7A*). Next, we inspected this phenomenon using electron microscopy (*Figure 7B*, *Figure 7—source data 1*). The presence of the R27 plasmid dictated the formation of clumps between the SL1344 strain and *E. coli* DH10BT1R (pNeae2 VHH-RSP #3). In contrast, no clumps or aggregates were observed when the plasmid-free SL1344 or the SL1344 (R27 Δrsp) strains were used.

To further characterize this clone, the corresponding VHH gene was cloned in the mammalian expression vector pIgΔCH1 (*Casasnovas et al., 2022*), which allows the fusion of the selected Nb domain to the Fc region of human IgG1. The corresponding Nb3-Fc protein fusion was purified from culture supernatants of transfected Expi293F cells (*Figure 8A*, *Figure 8—source data 1 and 2*), and its binding capacity to the RSP protein was analyzed by ELISA. The results indicated that this Nb3-Fc

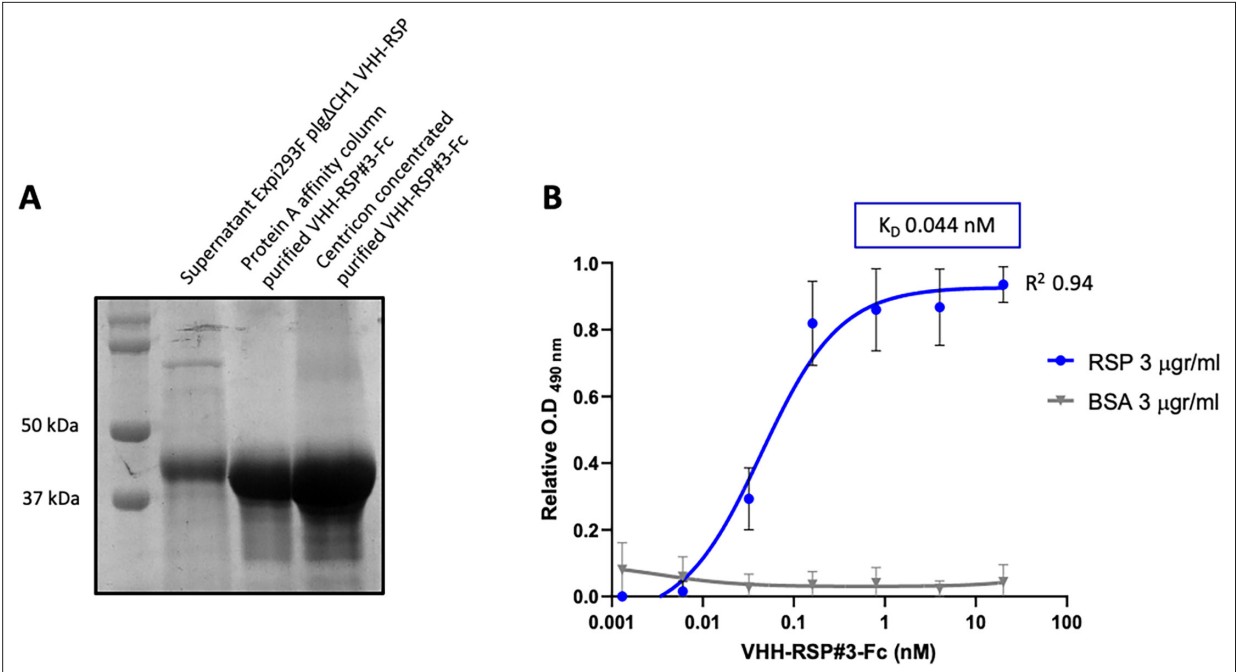

**Figure 8.** Characterization of the purified VHH-RSP#3 as a specific binder to the RSP protein. (**A**) Purification of Nb3-Fc by affinity column. Coomassie staining of the SDS-PAGE (10%) showing the purified Nb-RSP #3-Fc from transfected mammalian cell culture supernatants. Molecular weight markers are indicated on the left. (**B**) ELISA shows the binding capacity of the purified VHH-RSP#3-Fc to the RSP or BSA protein. The plot represents the OD values at 490 nm obtained with the indicated concentrations of the purified Nb3-Fc after 1 hr of interaction with the corresponding antigen.

The online version of this article includes the following source data for figure 8:

**Source data 1.** Original file for the Coomassie staining of the SDS-PAGE in *Figure 8A*.

**Source data 2.** PDF containing *Figure 8A* showing the Coomassie staining of the SDS-PAGE with highlighted sample labels corresponding to the purified Nb-RSP #3-Fc.

**Source data 3.** The Excel file containing numerical source data of the results is shown in *Figure 8B*.

presented a high affinity for binding to the RSP protein, showing an apparent equilibrium dissociation constant ($K_D$) in the picomolar range (~44 pM) (*Figure 8B*, *Figure 8—source data 3*).

## Interference of *E. coli* displaying Nb-RSP #3 strain on the conjugation of the R27 plasmid from the *Salmonella* Typhimurium SL1344 strain

Our objective was to investigate whether the *E. coli* surface expression of the Nb-RSP #3 would be able to interfere with the conjugation process of the R27 plasmid from the SL1344 *Salmonella* strain (donor) to the SL1344 *ibpA::lacZ-Km*[r] derivative (recipient). To prevent that the Nbs produced by the *E. coli* DH10BT1R (pNeae2 VHH-RSP #3) were titrated out by an excess of SL1344 (R27) donor cells, we planned a conjugation experiment with a low number of donor cells ($10^3$ cfu) mixed with an abundance of recipient cells (SL1344 *ibpA::lacZ-Km*[r]) ($10^6$ cfu), along with an excess of cells corresponding to the *E. coli* strain displaying Nb-RSP #3 ($10^7$ cfu).

Considering the known conjugation frequency of the R27 plasmid in *Salmonella* at 25 °C, which is approximately $10^{-3}$, and the low number of donor cells used in our experiment, we chose specific time intervals of 24, 36, and 48 hr following the mating process to detect transconjugants within the mating mixture. To accomplish this, the mating mixture was mixed with PB medium and incubated at 25 °C. To confirm that any effect on the conjugation frequency was solely due to the production of RSP-specific Nbs and not to other non-specific effects of Nb expression, we utilized as controls both a plasmid-free *E. coli* DH10BT1R strain and the same strain displaying an unrelated Nb binding human fibrinogen (*Salema et al., 2016*). The results obtained (*Figure 9*, *Figure 9—source data 1*) indicated that the surface display of Nb-RSP #3 by the interfering *E. coli* strain led to a significant and specific reduction in the conjugation frequency of the R27 plasmid in the *S.* Typhimurium SL1344 strain. Interfering *E. coli* strains with no Nb or a control Nb elicited a nonspecific ~10-fold reduction in the number of

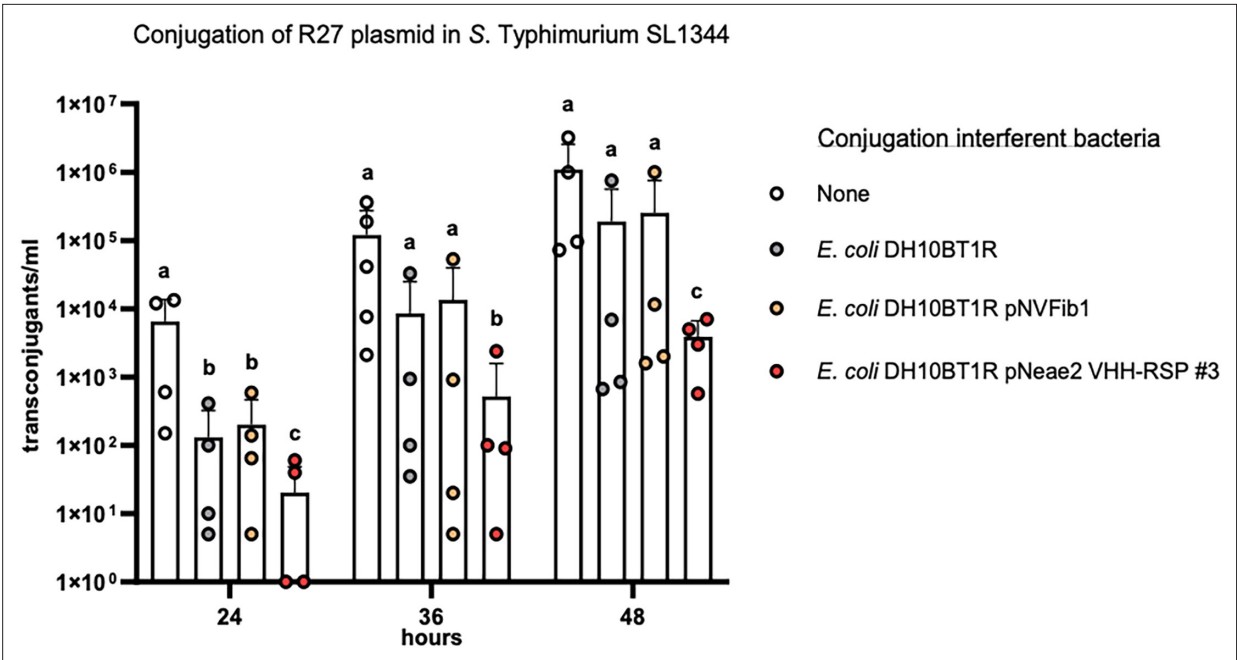

**Figure 9.** Interference of *E. coli* DH10BT1R cells expressing nanobodies on its surface with the conjugative transfer of the R27 plasmid in *Salmonella*. R27 plasmid was conjugated from a donor strain (SL1344) to a recipient strain (SL1344 *ibpA::lacZ-Km*') with the absence (labeled as 'none') or presence of the interferent strains *E. coli* DH10BT1R, plasmid-free clone; *E. coli* DH10BT1R (pNVFib1), clone expressing a fibrinogen-specific VHH; *E. coli* DH10BT1R (pNeae2 VHH-RSP #3), clone expressing RSP-specific VHH. Results are expressed as means ± SEMs (n=4 independent biological replicates). Means without a common letter differ, p<0.05. Data were analyzed with Student's t-test.

The online version of this article includes the following source data for figure 9:

**Source data 1.** The Excel file containing numerical source data of the results is shown in *Figure 9*.

transconjugants of R27 plasmid in all time points, compared to conjugation levels between *Salmonella* SL1344 strains. In contrast, interfering *E. coli* displaying Nb-RSP #3 reduced the number of transconjugants of R27 up to ~1000 fold at 24 hr and ~100 fold after 36 and 48 hr of mating (*Figure 9*). These results suggest a strong inhibitory effect in the conjugation of R27 due to the interaction between bacterial display Nb-RSP #3 and RSP protein expressed by donor cells. To rule out that the observed reduction in the conjugation frequency in mixtures containing the *E. coli* strain DH10BT1R (pNeae2 VHH-RSP #3) and the SL1344 (R27) strains could be due to the specific transfer of the R27 plasmid to the interferent *E. coli* DH10BT1R (pNeae2 VHH-RSP #3) strain, we also detected transconjugants of the *E. coli* DH10BT1R (pNeae2 VHH-RSP #3) and *E. coli* DH10BT1R (pNVFib1) interferent strains by plating them on LB medium containing Tc (R27 plasmid marker) and Cm (pNeae2 plasmid marker). In both experiments, the conjugation frequency was the same (1.5×10$^{-2}$ transconjugants/donor cells).

## Discussion

Vaccination stands as a priority strategy in addressing the challenges posed by AMR (*Frost et al., 2022*; *Micoli et al., 2021*). Vaccines of special interest should target the agents responsible for significant health care-associated infections, which include multiple resistant Gram-negative bacteria (*Lipsitch and Siber, 2016*). Protecting against specific pathogens (i.e. *S. pneumoniae, K. pneumoniae*) reduced the incidence of infections caused by them, leading to a decrease in antibiotic usage and hence reducing the proliferation of AMR strains (*Buchy et al., 2020*).

One of the challenges in vaccine development is the identification of broadly protective antigens (*Troisi et al., 2020*). In the context of AMR, current strategies focus on targeting either the most common antigens among AMR clones of a specific pathogen (*Micoli et al., 2021*) or specific virulence determinants (*Moriel et al., 2010*; *Nesta et al., 2012*). However, an alternative approach is to develop a vaccine that directly targets a resistance determinant, which would be effective not only against a specific microorganism but also against any microorganism expressing the corresponding

resistance determinant. Examples of successful targeting include penicillin-binding proteins or beta-lactamases, which have demonstrated protection against infections caused by *Staphylococcus aureus* (*Senna et al., 2003*), *Neisseria meningitidis* (*Zarantonelli et al., 2006*), or *Pseudomonas aeruginosa* (*Ciofu et al., 2002*).

The novelty of the approach presented in this work is that, instead of targeting an antigen involved in an antibiotic resistance mechanism, we focus on an antigen associated with the transmission of resistance determinants.

Big proteins have been previously shown to be good vaccine candidates (*Khanum et al., 2022*; *Koizumi and Watanabe, 2004*). The antigenicity of the RSP protein has been demonstrated by the presence of higher levels of specific immunoglobulins. Immunized mice exhibited higher levels of IgG in serum and IgA in the colon, jejunum, and feces compared to control mice. In terms of the protective effect of immunization with the RSP protein in mice, the results can be interpreted as follows. The secretion of proinflammatory cytokines in the colon was comparable between immunized and non-immunized animals, as well as the reduction in body weight. However, clinical symptoms, *Salmonella* counts in the spleen, and the weight and secretion of proinflammatory cytokines in this organ were significantly reduced in immunized animals. While immunization with the RSP protein did not provide complete protection against intestinal infection with the SL1344 strain, it did result in attenuated disease symptoms and restricted systemic infection. These findings support further research to validate the RSP protein as a valuable antigen for protection against bacterial infections caused by enteric bacteria harboring IncHI plasmids.

IncHI plasmids are commonly found in AMR bacteria isolated from clinical samples (*Parvez and Khan, 2018*; *Zhang et al., 2021*). They are also frequently identified in *Salmonella* isolates (*Phan and Wain, 2008*). The emergence of AMR *S. typhi* isolates has been reported in developing countries (*Tanmoy et al., 2018*). Developing an effective vaccine against this pathogen would be particularly beneficial for children in endemic countries, including regions such as South Asia, Southeast Asia, and sub-Saharan Africa, as well as for travelers (*World Health Organization, 2019*). The RSP protein shows promise as a valuable antigen to be included in such a vaccine, offering potential protection against AMR *Salmonella* infections associated with IncHI plasmids.

An alternative approach to provide protection against infections caused by virulent AMR strains carrying IncHI plasmids is to specifically target the RSP protein with antibodies. Monoclonal antibodies against various bacterial pathogens have demonstrated effectiveness, with some currently in advanced stages of clinical trials (*Cook and Wright, 2022*). Nanobodies, a novel class of antibodies, offer a wide range of clinical applications (*Graf et al., 2019*; *Jovčevska and Muyldermans, 2020*). The use of nanobodies specifically targeting the RSP protein represents a promising approach to combat infections caused by pathogenic bacteria harboring IncHI plasmids. Inhibiting the transfer of antimicrobial resistance plasmids has been identified as a crucial strategy to combat AMR (*Buckner et al., 2018*; *Graf et al., 2019*). Our study demonstrates that *E. coli* strains producing RSP-specific nanobodies can bind to *Salmonella* cells carrying the R27 plasmid, resulting in a reduction in the conjugation frequency.

These results reinforce the notion that expressing RSP-specific nanobodies in a probiotic bacterium, such as *E. coli* Nissle 1917, could potentially limit the transfer of IncHI plasmids in natural settings, such as livestock farms. Furthermore, RSP-specific antibodies could be utilized to combat infections caused by pathogenic isolates carrying IncHI plasmids. Nanobodies, known for their heat- and acid-stability and solubility, show potential for oral administration to control gastrointestinal infections (*Yuki et al., 2023*). Thus, either purified RSP-specific nanobodies or probiotic microbial cells expressing these nanobodies could be orally administered to combat infections caused by AMR virulent strains harboring IncHI plasmids.

The expression of plasmid-encoded Big proteins is not exclusive to IncHI plasmids; other plasmids such as IncA/C and IncP2 also encode these proteins (*Hüttener et al., 2022*). IncA/C plasmids were initially identified in the 1970s among multidrug-resistant *Aeromonas hydrophila* and *Vibrio* strains that infected cultured fish (*Aoki et al., 1971*). Since the 1990s, IncA/C plasmids have gained significant attention due to their ability to mobilize antimicrobial resistance determinants in enterobacteria and other Gram-negative microorganisms (*Call et al., 2010*; *Fernández-Alarcón et al., 2011*; *Fricke et al., 2009*; *Welch et al., 2007*). These plasmids have an exceptionally wide host range, encompassing Beta-, Gamma-, and Deltaproteobacteria (*Suzuki et al., 2010*), and they play a crucial role in

the global dissemination of antimicrobial resistance (*Eda et al., 2020*; *Rozwandowicz et al., 2018*). In a recent study, IncA/C plasmids accounted for 50% of all plasmids isolated from clinical *K. pneumoniae* strains harboring the *blaNDM* gene (*Qamar et al., 2021*). Notably, the monophasic variant of *S.* Typhimurium, *S. enterica* serovar 4,[5],12:i:-, which has emerged as a global cause of multidrug-resistant salmonellosis, predominantly harbors IncHI1 and IncA/C resistance plasmids (*Ingle et al., 2021*).

Therefore, targeting these plasmid-encoded Big proteins through immunotherapies can be a valuable strategy to combat infections caused by pathogenic microorganisms carrying various types of resistance plasmids.

## Methods

### Bacterial strains and growth conditions

The bacterial strains (see *Supplementary file 1a*) were routinely grown in Luria-Bertani (LB) medium (10 g/L NaCl, 10 g/L tryptone, and 5 g/L yeast extract) with vigorous shaking at 200 rpm (Innova 3100; New Brunswick Scientific). The antibiotics used were chloramphenicol (Cm) (25 µg/mL), tetracycline (Tc) (15 µg/mL), carbenicillin (Cb) (100 µg/mL), and kanamycin (Km) (50 µg/mL) (Sigma-Aldrich). *E. coli* strains carrying the plasmid with a VHH gene were grown at 30 °C on LB-agar plates (1.5% w/v) or in LB liquid medium with the addition of chloramphenicol (Cm, 30 µg/mL) for plasmid selection. To overexpress the nanobodies in the bacterial surface of *E. coli* DH10BT1R, firstly bacteria were grown overnight (O/N) under statics conditions in LB medium containing 2% (w/v) of glucose. Then, bacteria were harvested by centrifugation (4000 g for 5 min) and grown in the same media supplemented with 0.05 mM isopropylthio-β-D-galactoside (IPTG) in constant agitation (160 rpm) for 16 hr.

### Plasmids

The different plasmids used in this work are listed in *Supplementary file 1a*.

### Oligonucleotides

The oligonucleotides used in this work are listed in *Supplementary file 1b*.

### Genetic manipulations

All enzymes used to perform standard molecular and genetic procedures were used according to the manufacturer's recommendations. To introduce plasmids into *E. coli*, *Salmonella*, and *Klebsiella*, bacterial cells were grown until an O.D$_{600\,nm}$ of 0.8. Cells were then washed several times with 10% glycerol, and the respective plasmids or DNA were electroporated by using an Eppendorf gene pulser (Electroporator 2510).

Deletion of the *flhDC* locus was performed in the strain SL1344 by using the $\lambda$ Red recombination method, as previously described (*Datsenko and Wanner, 2000*).

### Conjugative transfer of the R27 plasmid

The R27 plasmid was conjugated as described previously (*Hüttener et al., 2018*). The mating frequency was calculated as the number of transconjugants per donor cell.

### Purification of the native RSP protein for the immunization assay

The RSP protein was purified from cell-free supernatant fractions obtained from the *Salmonella* cells lacking the flagella (SL1344 R27 *flhDC::Km*). Cells were grown at 25 °C in LB medium for 16 hr in constant shaking at 200 rpm, then centrifugated at 7000 rpm for 30 min at room temperature. Cell pellets were discarded, and the supernatant was filtered through a 0.22 µm filter (Millipore). Supernatants were then ultra-centrifuged at 40,000 rpm for 1 hr at 4 °C and carefully aspired to not disturb the protein pellet and discarded. The remaining pellet was solubilized using commercial Phosphate-buffered saline (PBS – Gibco). Proteins were quantified using the Bradford standard method (Pierce).

### Purification of the C-terminal domain of the RSP protein

For the RSP protein production, the carboxyl-terminal region of the RSP protein was amplified (280 AA in total). Amplification of that region was achieved by performing PCR using the R27 plasmid

as a DNA template and the primers RSP5 BamHI Fw and RSP PstI Rv, together with the Phusion Hot Start II High-fidelity DNA Polymerase (Thermo Fisher Scientific) following the manufacturer's recommendations. The DNA was then purified using the GeneJet PCR Purification Kit (Thermo Fisher Scientific), digested with BamHI and PstI restriction endonucleases according to the manufacturer´s instructions (New England Biolabs), and ligated into the pMAL-p2E vector digested with the same restriction endonucleases using the T4 DNA Ligase (New England Biolabs) according to the manufacturer´s instructions. The resulting plasmid, termed pMAL-RSP#5/7, was Sanger sequenced and transformed into BL21 cells. Cells transformed with pMAL-RSP#5/7 plasmid were grown in LB medium supplemented with carbenicillin at a final concentration of 100 µg/mL at 37 °C for 16 hr at constant agitation at 200 rpm. Then, cells were diluted 1:100 in LB with 100 µg/mL of carbenicillin and glucose at 0.45% of the final concentration. Cells were incubated at 37 °C with 200 rpm until reached the $O.D_{600\,nm}$ of 0.5, then IPTG was added at a final concentration of 0.15 mM. Cells were incubated at 37 °C for 2 hr under constant agitation of 200 rpm for the induction of the recombinant protein. Cells were then centrifuged at 7500 rpm for 30 min at 4 °C. The cells pellet was subsequently resuspended in column buffer (20 mM HEPES (Na) pH 7.5, 200 mM NaCl, 1 mM EDTA, 1 mM β-mercaptoethanol) plus protease inhibitor (Complete Ultra Tablets, Mini, EDTA-free, EASYpack, Roche). Cells were then disrupted by French press cell lysis and centrifugated at 12,000 rpm for 1 hr at 4 °C. Supernatants were filtered through a 0.22 µM filter (Millipore) and mixed with 250 µl of Amylose resin (New England Biolabs) for 2 hr at 4 °C in an orbital shaker. The resin was then loaded in chromatography columns and washed twice with the column buffer. The recombinant RSP C-terminal protein was then eluted using the elution buffer (20 mM HEPES (Na) pH 7.5, 200 mM NaCl, 1 mM EDTA, 1 mM β-mercaptoethanol and 10 mM maltose). Eluted fractions were collected and then concentrated using Amicon Ultra-15 Ultracel 10 K (Millipore) according to the manufacturer´s instructions.

## Immunization of dromedaries with the C-terminal fragment of the RSP protein and selection of clones producing anti-RSP specific nanobodies

Two different dromedary camels (*Camelus dromedarious*) were immunized with purified RSP protein (~1.2 mg) mixed with veterinary vaccine adjuvant (GERBU). After six rounds of inoculations by subcutaneous injections, serial dilutions of pre-immune and immune sera were used in ELISA to confirm the immune response of dromedary after the RSP protein inoculation. B lymphocytes from peripheral blood were isolated, RNA was extracted and VHH genes were amplified by RT-PCR using the VHH Sfi and VHH Not oligonucleotides following the protocol described previously (*Salema et al., 2013*).

The amplified VHH fragments were digested with SfiI and NotI restriction enzymes and ligated into the pNeae2 backbone vector digested with the same restriction enzymes, and finally transformed in *E. coli* DH10BT1R by electroporation. Following this protocol, a library of 8.75×10⁷ clones was obtained.

This library was used for the enrichment of clones that recognized the RSP protein. To this end, purified RSP protein was mixed with Biotin-NHS (Biotinamidocaproate N-hydroxysuccinimide ester, Sigma-Aldrich) and three rounds of MACS (with RSP concentrations of 100 nM the first one) and 50 nM the other two were performed. Thereafter, a FACS selection with a protein concentration of 50 nM were performed. The positive clones for the RSP binding were selected for further analysis.

## Cloning and purification of the RSP-specific nanobody (VHH-RSP #3)

The gene coding for the RSP-specific selected Nb was PCR amplified with oligonucleotides VHH pIg AgeI and VHH pIg BamHI using the pNeae2 VHH-RSP #3 purified plasmid as a template. This PCR product was subsequently cloned in AgeI-BamHI sites of the mammalian expression vector pIgΔCH1 (*Casasnovas et al., 2022*) derived from pIgγ1HC (*Tiller et al., 2008*). This vector allows the expression of the cloned gene located in the frame with an IgH signal peptide fused to the human IgG1 hinge and Fc portion. For the nanobody overexpression and purification, the Expi293 Expression System Kit (Thermo Fisher Scientific) was used. The mammalian cells transfected with the purified expression vector (pIgΔCH1 VHH-RSP) were grown 5 days post-transfection. Cells were then centrifuged, the supernatant was collected and filtered, and the proteins were purified with an Ig Select or protein A affinity column (Cytiva), following the manufacturer's instructions.

## Enzyme-linked immunosorbent assays to evaluate the specific VHH-RSP #3 binding to the RSP protein

ELISA experiments were performed in 96-well plates (Maxisorp, Nunc), which were previously coated overnight at 4 °C with 50 μl/well of the purified RSP protein antigen or BSA at 3 μg/ml in PBS. Antigen-coated plates were washed with PBS and blocked with 200 μl/well of 3% (w/v) skimmed milk in PBS at room temperature for 2 hr. Afterwards, different dilutions of the VHH-RSP (prepared in 3% (w/v) skimmed milk in PBS) were added to the wells for 1 hr. After the incubation, the wells were washed three times with PBS, and rabbit anti-human Fc IgG-HRP (Jackson ImmunoResearch) was added to the plates (diluted 1:5000 in 3% (w/v) skimmed milk in PBS) to detect the bound Nbs. After 1 hr of incubation at room temperature, plates were washed with PBS and developed with $H_2O_2$ and o-phenylenediamine (OPD; Merck-Sigma). The Optical Density (OD) at 490 nm ($OD_{490\,nm}$) of plate wells was determined (iMark, Bio-Rad), and the values were corrected with the background levels of $OD_{490nm}$ of the wells without antigen added.

## In vivo immunization and infection of mice

Male and female C57BL/6 mice were purchased from Envigo (Bresso, Italy) and maintained under stable temperature and humidity conditions with a 12 hr light and 12 hr dark cycle and free access to food and water.

### Set up of the infection protocol with the SL1344 (pHCM1) strain

To induce infection, the *S.* Typhimurium strain SL1344 (pHCM1) was chosen. An assay was designed to assess whether infection with this strain is similar to infection with the plasmid-free strain. Four groups of mice were infected with strains SL1344 and SL1344 (pHCM1), and two of them were treated with 150 mg/kg/day ampicillin (Amp) (Sigma-Aldrich) in drinking water and two of them were untreated. In addition, a group of uninfected control mice was included as a reference group. Thus, the groups were: (i) Control: uninfected mice; (ii) Sal: SL1344-infected mice not treated with Amp; (iii) Sal-pHCM1: SL1344-pHCM1-infected mice not treated with Amp; (iv) Sal WT +Amp: SL1344-infected mice treated with Amp; (v) Sal pHCM1 +Amp: SL1344-pHCM1-infected mice treated with Amp. Three hours before the challenge with the bacterial suspension, mice received an intraperitoneal injection of cimetidine (50 mg/kg; Sigma-Aldrich) to reduce acid secretion and improve bacterial survival (*Ren et al., 2014*). To induce infection, 100 μl of the bacterial suspension ($10^9$ cfu of *Salmonella*) was administered by oral gavage. During the infection period, the body weight of animals was monitored daily. Four to six days after infection (depending on survival), the animals were evaluated for clinical signs of disease, and fecal samples were obtained. They were subsequently euthanized and samples of spleen, colon mucosa, intestinal lavage, and serum were obtained. Samples were obtained and processed as described previously (*Miró et al., 2023*).

The parameters analyzed to score the clinical signs were: Coat care (normal, 0; slightly altered, 1; considerably altered, 2; very altered, 3); posture (normal, 0; slightly curved, 1; considerably curved, 2; very curved, 3); movement (normal, 0; slightly slow, 1; considerably slow, 2; very slow, 3); defecation (normal, 0; soft stools, 1; watery stools, 2; liquid stools, 3); body weight loss (no weight loss, 0; weight loss less than 5%, 1; weight loss between 5 and 10%, 2; weight loss greater than 10%, 3).

### Immunization with the RSP protein and infection with the *S.* Typhimurium SL1344 (pHCM1) strain

Regarding the adjustment of the immunization with the RSP protein as antigen, we proceeded as previously described (*Miró et al., 2023*). Briefly, mice were intranasally administered with 3 μg of the RSP protein and 5 μg of cholera toxin (CT; Sigma-Aldrich) as an immunological adjuvant. Immunization was performed three times every 2 weeks (at 3, 5, and 7 weeks of age). Non-immunized mice received CT alone. Mice were distributed randomly into two groups: Non-Imm (Non-immunized, CT alone); Imm (Immunized mice). During the immunization period, the body weight of animals was monitored weekly. Two weeks after the last immunization, animals were euthanized and samples of feces, intestinal lavage, and serum were obtained. Total and RSP-specific immunoglobulins were determined as explained in the section Immunoglobulin determination.

In order to evaluate the protective efficacy of immunization, the previously established infection protocol with the *S. Typhimurium* SL1344 (pHCM1) strain was used. The experiments were performed using a Latin design with two factors: immunization of mice using the RSP protein as immunogen; and subsequently infected with the *S. Typhimurium* SL1344 (pHCM1) strain. The animals were randomly distributed into four groups: (i) non-immunized and non-infected mice (Non-Imm); (ii) non-immunized and *Salmonella*-infected animals (Sal); (iii) immunized and non-infected animals with *Salmonella* (Imm); (iv) immunized and infected animals (Imm-Sal). The infection was provoked two weeks after the last immunization dose.

## Bacterial counts in the spleen

Spleen samples were homogenized with sterile PBS (20 mg/mL) using a PRO200 homogenizer (Pro-Scientific, USA) at 18,000 g, and homogenates were used to count the presence of *Salmonella* cells plated in MacConkey media agar plates.

## Immunoglobulin concentration

Total secreted IgA and anti-RSP-specific IgA were determined in intestinal and colon content and in feces by sandwich enzyme-linked immunosorbent assay (ELISA) as previously described (*Ren et al., 2014*). Briefly, plates were coated with anti-mouse IgA monoclonal antibody (1 μg/mL; Sigma-Aldrich) in PBS to quantify total IgA concentration or RSP (20 μg/mL) in PBS to measure RSP-specific IgA concentration. Mouse IgA (Bethyl) was used as a standard for the determination of total IgA. Goat anti-mouse IgA (Bethyl) conjugated with horseradish peroxidase (HRP) was used as the detection antibody.

The concentrations of total IgG and specific IgG against the RSP protein in serum were also determined. Plates were coated with goat anti-mouse IgG (1 μg/mL; Sigma-Aldrich) to quantify the concentration of total IgG or RSP (10 μg/mL) in PBS to measure the concentration of specific IgG against RSP. Mouse IgG (Sigma-Aldrich) was used as a standard for the determination of total IgG. Goat anti-mouse IgG conjugated with HRP (Sigma-Aldrich) was used as the detection antibody.

In all cases, o-phenylenediamine (OPD, 0.4 mg/mL; Sigma-Aldrich) was used as HRP substrate, and the color intensity was measured at 492 nm in a microplate reader (Sunrise).

## Cytokine expression in spleen and colon

RNA extraction and reverse transcription were carried out as previously described (*Rosell-Cardona et al., 2022*). RNA quality and quantity were assessed by spectrophotometry (NanoDrop ND-1000; Thermo Fisher Scientific) and its integrity was determined with an Agilent 2100 Bioanalyzer (Agilent Technologies Inc). In all cases, the RNA integrity was ≥9 and the A260/280 ratio was between 1.96 and 2.02. Total RNA was reverse-transcribed using an iScript cDNA Synthesis Kit (Bio-Rad). For real-time PCR determinations, we used SsoAdvanced Universal SYBR Green Supermix (Bio-Rad). The primers used are described previously (*Garcia-Just et al., 2020*). Real-time PCR was performed on a MiniOpticon Real-Time PCR System (Bio-Rad). Each PCR run included duplicates of reverse transcript cDNA for each sample and negative controls (reverse transcription-free samples, RNA-free samples). Quantification of the target gene transcripts was conducted using hypoxanthine phosphoribosyl transferase 1 (*hprt1*) gene expression as a reference and was performed with the $2^{-\Delta\Delta CT}$ method (*Schmittgen and Livak, 2008*). Product fidelity was confirmed by melting curve analysis.

## Immunogold electron microscopy

Immunogold microscopy experiments were performed as previously described (*Hüttener et al., 2019*).

## Statistical analysis

The results of mice experiments are presented as mean ± SEM, except for survival curves and disease indices. Survival curves have been compared by the Log-rank (Mantel-Cox) test and expressed as a percentage of survival. Disease indices are expressed as median values and quartiles and have been compared by the Kruskal-Wallis test. Body weight evolution was analyzed by means of repeated measures ANOVA. Continuous data were analyzed with Levene's test to assess the homogeneity of variance and with the Shapiro-Wilk test to assess distribution. Homogeneous and normally distributed

data were analyzed with Student's t-test (immunization-tuning experiments) or one-way analysis of variance (ANOVA) (infection-tuning) followed by Fisher's least significant difference (LSD) post hoc test and two-way ANOVA (immunization and infection experiments). Student's t-test was also used to determine statistical the significance of R27 conjugation frequency in mating experiments. Data were analyzed using GraphPad Prism software v.9.3.1 (GraphPad Software, Inc). Differences were considered significant at $p < 0.05$.

## Acknowledgements

This work was supported by grants from Fundació 'La Marató TV3,' Spain (project 201818 10), CERCA Program/Generalitat de Catalunya to AJ, PID2021-124676NB-I00 from the Ministerio de Economía, Industria y Competitividad to SM and JT. Work in the laboratory of LAF is supported by grants PLEC2021-007739 from MCIN/AEI NextGeneration EU/ PRTR and FET Open 965018-BIOCELLPHE from the European Union's Horizon 2020 Future and Emerging Technologies research and innovation program. LM and APB are members of the Institut de Recerca en Nutrició, i Seguretat Alimentària (INSA-UB), which is recognized as a Maria de Maeztu Unit of Excellence and funded by MICIN/AEI/ FEDER (CEX2021-001234-M).

## Additional information

### Funding

| Funder | Grant reference number | Author |
|---|---|---|
| Fundació la Marató de TV3 | 20181810 | Antonio Juarez |
| Ministerio de Economía y Competitividad | PID2021-124676NB-I00 | Susana Merino Joan Tomas |
| Ministerio de Ciencia e Innovación | PLEC2021-007739 | Luis Ángel Fernández |
| Horizon 2020 Framework Programme | 965018-BIOCELLPHE | Luis Ángel Fernández |
| Ministerio de Ciencia e Innovación | CEX2021-001234-M | Anna Perez-Bosque |

The funders had no role in study design, data collection and interpretation, or the decision to submit the work for publication.

### Author contributions

Alejandro Prieto, Anna Perez-Bosque, Data curation, Formal analysis, Investigation, Writing - original draft, Writing – review and editing; Luïsa Miró, Data curation, Software, Formal analysis, Investigation, Writing – review and editing; Yago Margolles, Manuel Bernabeu, David Salguero, Susana Merino, Joan Tomas, Juan Alberto Corbera, Investigation; Mario Huttener, Luis Ángel Fernández, Investigation, Writing – review and editing; Antonio Juarez, Conceptualization, Supervision, Validation, Investigation, Writing - original draft, Project administration, Writing – review and editing

### Author ORCIDs

Alejandro Prieto http://orcid.org/0000-0002-9175-5809
Juan Alberto Corbera https://orcid.org/0000-0001-7812-2065
Anna Perez-Bosque http://orcid.org/0000-0003-2175-9684
Antonio Juarez https://orcid.org/0000-0002-5027-8778

### Ethics

EthicsThe dromedary camel immunization protocol followed European Union and Spanish guidelines of animal experimentation and was approved by the Ethics Committee for Animal Experimentation from University of Las Palmas de Gran Canaria and the Consejería de Agricultura, Pesca y Aguas of the Canary Islands Goverment (reference OEBA_ULPGC_20–2023). Experimentation with mice was conducted at Animal Facility of Faculty of Pharmacy and Food Science. Experiments were approved

by the Ethics Committee of Animal Experimentation of the University of Barcelona (Ref. 170/20). Animals were handled in strict accordance with the guidelines of the European Community 86/609/CEE.

Reviewer #1 (Public review): https://doi.org/10.7554/eLife.95328.3.sa1
Reviewer #2 (Public review): https://doi.org/10.7554/eLife.95328.3.sa2
Author response https://doi.org/10.7554/eLife.95328.3.sa3

## Additional files

### Supplementary files
• Supplementary file 1. Strains, plasmids and oligonucleotides used in this work. (**a**) List of the strains and plasmids used in this work. (**b**) List of the oligonucleotides used in this work.
• MDAR checklist

### Data availability
All data generated or analysed during this study are included in the manuscript supporting files.

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
