## [Editor Report · eLife assessment]

This **important** and novel study addresses the challenge of antimicrobial resistance by targeting plasmid proteins that interfere with plasmid transfer as a strategy to limit the spread of antibiotic-resistance genes. The evidence presented and the integration of two approaches to tackle antimicrobial resistance is **convincing**. This work will interest those working on plasmid transfer and antimicrobial resistance.

---

## [Referee Report · Reviewer #1 (Public review)]

The study by Prieto et al. faces the increasingly serious problem of bacterial resistance to antimicrobial agents. This work has an important element of novelty proposing a new approach to control antibiotic resistance spread by plasmids. Instead of targeting the resistance determinant, plasmid-borne proteins are used as antigens to be bound by specific nanobodies (Nbs). Once bound plasmid transfer was inhibited and *Salmonella* infection blocked. This in-depth study is quite detailed and complex, with many experiments (9 figures with multiple panels), rigorously carried out. Results fully support the authors' conclusions. Specifically, the authors investigated the role of two large molecular weight proteins (RSP and RSP2) encoded by the IncHI1 derivative-plasmid R27 of *Salmonella*. These proteins have bacterial Ig-like (Big) domains and are expressed on the cell surface, creating the opportunity for them to serve as immunostimulatory antigens. Using a mouse infection model, the authors showed that RSP proteins can properly function as antigens, in *Salmonella* strains harboring the IncHI1 plasmid. The authors clearly showed increased levels of specific IgG and IgA antibodies against these RSP proteins proteins in different tissues of immunized animals. In addition, non-immunized mice exhibited *Salmonella* colonization in the spleen and much more severe disease than immunized ones.

However, the strength of this work is the selection and production of nanobodies (Nbs) that specifically interact with the extracellular domain of RSP proteins. The procedure to obtain Nbs is lengthy and complicated and includes the immunization of dromedaries with purified RPS and the construction of a VHH (H-chain antibody variable region) library in *E. coli*. As RSP is expressed on the surface of *E. coli,* specific Nbs were able to agglutinate *Salmonella* strains harboring the p27 plasmid encoding the RSP proteins.

The authors demonstrated that Nbs-RSP reduced the conjugation frequency of p27 thus limiting the diffusion of the amp resistance harbored by the plasmid. This represents an innovative and promising strategy to fight antibiotic resistance, as it is not blocked by the mechanism that determines, in the specific case, the amp resistance of p27 but it targets an antigen associated with HincHI- derivative plasmids. Thus, RPS vaccination could be effective not only against *Salmonella* but also against other enteric bacteria. A possible criticism could be that Nbs against RSP proteins reduce the severity of the disease but do not completely prevent the infection by *Salmonella*.

---

## [Referee Report · Reviewer #2 (Public review)]

Summary:

This manuscript aims to tackle the antimicrobial resistance through the development of vaccines. Specifically, the authors test the potential of the RSP protein as a vaccine candidate. The RSP protein contains bacterial Ig-like domains that are typically carried in IncHl1 plasmids like R27. The extracellular location of the RSP protein and its role in the conjugation process makes it a good candidate for a vaccine. The authors then use *Salmonella* carrying an IncHl plasmid to test the efficacy of the RSP protein as a vaccine antigen in providing protection against infection of antibiotic-resistant bacteria carrying the IncHl plasmid. The authors found no differences in total IgG or IgA levels, nor in pro-inflammatory cytokines between immunized and non-immunized mice. They however found differences in specific IgG and IgA, attenuated disease symptoms, and restricted systemic infection.

The manuscript also evaluates the potential use of nanobodies specifically targeting the RSP protein by expressing it in *E. coli* and evaluating their interference in the conjugation of IncHl plasmids. The authors found that *E. coli* strains expressing RSP-specific nanobodies bind to *Salmonella* cells carrying the R27 plasmid thereby reducing the conjugation efficacy of *Salmonella*.

Strengths:

- The main strength of this manuscript is that it targets the mechanism of transmission of resistance genes carried by any bacterial species, thus making it broad.

- The experimental setup is sound and with proper replication.

Weaknesses:

- The two main experiments, evaluating the potential of the RSP protein and the effects of nanobodies on conjugation, seem as parts of two different and unrelated strategies.

- The survival rates shown in Figure 1A and Figure 3A for *Salmonella* pHCM1 and non-immunized mice challenged with *Salmonella*, respectively, are substantially different. In the same figures, the challenge of immunized mice and *Salmonella* pHCM1 and mice challenged with *Salmonella* pHCM1 with and without ampicillin are virtually the same. While this is not the only measure of the effect of immunization, the inconsistencies in the resulting survival curves should be addressed by the authors more thoroughly as they can confound the effects found in other parameters, including total and specific IgG and IgA, and pro-inflammatory cytokines.

- Overall the results are inconsistent and provide only partial evidence of the effectiveness of the RSP protein as a vaccine target.

- The conjugative experiments use very long conjugation times, making it harder to asses if the resulting transconjugants are the direct result of conjugation or just the growth of transconjugants obtained at earlier points in time. While this could be assessed from the obtained results, it is not a direct or precise measure.

- While the potential outcomes of these experiments could be applied to any bacterial species carrying this type of plasmids, it is unclear why the authors use *Salmonella* strains to evaluate it. The introduction does a great job of explaining the importance of these plasmids but falls short in introducing their relevance in *Salmonella*.

---

## [Author Response]

The following is the authors’ response to the original reviews.

**Public Reviews:**

**Reviewer #1 (Public Review):**
The study by Prieto et al. faces the increasingly serious problem of bacterial resistance to antimicrobial agents. This work has an important element of novelty proposing a new approach to control antibiotic resistance spread by plasmids. Instead of targeting the resistance determinant, plasmid-borne proteins are used as antigens to be bound by specific nanobodies (Nbs). Once bound plasmid transfer was inhibited and *Salmonella* infection blocked. This in-depth study is quite detailed and complex, with many experiments (9 figures with multiple panels), rigorously carried out. Results fully support the authors' conclusions. Specifically, the authors investigated the role of two large molecular weight proteins (RSP and RSP2) encoded by the IncHI1 derivative-plasmid R27 of *Salmonella*. These proteins have bacterial Ig-like (Big) domains and are expressed on the cell surface, creating the opportunity for them to serve as immunostimulatory antigens. Using a mouse infection model, the authors showed that RSP proteins can properly function as antigens, in *Salmonella* strains harboring the IncHI1 plasmid. The authors clearly showed increased levels of specific IgG and IgA antibodies against these RSP proteins proteins in different tissues of immunized animals. In addition, non-immunized mice exhibited *Salmonella* colonization in the spleen and much more severe disease than immunized ones.However, the strength of this work is the selection and production of nanobodies (Nbs) that specifically interact with the extracellular domain of RSP proteins. The procedure to obtain Nbs is lengthy and complicated and includes the immunization of dromedaries with purified RPS and the construction of a VHH (H-chain antibody variable region) library in *E. coli.* As RSP is expressed on the surface of *E. coli*, specific Nbs were able to agglutinate *Salmonella* strains harboring the p27 plasmid encoding the RSP proteins.The authors demonstrated that Nbs-RSP reduced the conjugation frequency of p27 thus limiting the diffusion of the amp resistance harbored by the plasmid. This represents an innovative and promising strategy to fight antibiotic resistance, as it is not blocked by the mechanism that determines, in the specific case, the amp resistance of p27 but it targets an antigen associated with HincHI- derivative plasmids. Thus, RPS vaccination could be effective not only against *Salmonella* but also against other enteric bacteria. A possible criticism could be that Nbs against RSP proteins reduce the severity of the disease but do not completely prevent the infection by *Salmonella*.

It is true that vaccina2on of mice with purified RSP protein did not provide complete protec2on against infec2on with a *Salmonella* strain harboring an IncHI plasmid. As this finding is based on an animal model, further inves2ga2on is required to evaluate its clinical efficacy. In any case, even par2al protec2on provided by nanobodies or by a vaccine could poten2ally improve survival rates among cri2cally ill pa2ents infected with a pathogenic bacterium harboring an IncHI plasmid. An addi2onal beneficial aspect of our approach is that it will reduce dissemina2on of IncHI plasmids among pathogenic bacteria, which would reduce the presence of an2bio2c resistance plasmids in the environment and in the bacteria infec2ng pa2ents.

**Reviewer #2 (Public Review):**
Summary:This manuscript aims to tackle the antimicrobial resistance through the development of vaccines. Specifically, the authors test the potential of the RSP protein as a vaccine candidate. The RSP protein contains bacterial Ig-like domains that are typically carried in IncHl1 plasmids like R27. The extracellular location of the RSP protein and its role in the conjugation process makes it a good candidate for a vaccine. The authors then use *Salmonella* carrying an IncHl plasmid to test the efficacy of the RSP protein as a vaccine antigen in providing protection against infection of antibioticresistant bacteria carrying the IncHl plasmid. The authors found no differences in total IgG or IgA levels, nor in pro-inflammatory cytokines between immunized and non-immunized mice. They however found differences in specific IgG and IgA, attenuated disease symptoms, and restricted systemic infection.The manuscript also evaluates the potential use of nanobodies specifically targeting the RSP protein by expressing it in *E. coli* and evaluating their interference in the conjugation of IncHl plasmids. The authors found that *E. coli* strains expressing RSPspecific nanobodies bind to *Salmonella* cells carrying the R27 plasmid thereby reducing the conjugation efficacy of *Salmonella*.Strengths:The main strength of this manuscript is that it targets the mechanism of transmission of resistance genes carried by any bacterial species, thus making it broad.The experimental setup is sound and with proper replication.Weaknesses:The two main experiments, evaluating the potential of the RSP protein and the effects of nanobodies on conjugation, seem as parts of two different and unrelated strategies.

In preparing our manuscript, we were aware that we included two different strategies to combat an2microbial resistance. However, we deemed it valuable to include both in the paper. The development of new vaccines and the inhibi2on of the transfer of an2bio2c resistance determinants are currently considered relevant approaches to combat an2microbial resistance. Our inten2on in the ar2cle is to integrate these two strategies.

The survival rates shown in Figure 1A and Figure 3A for *Salmonella* pHCM1 and non-immunized mice challenged with *Salmonella*, respectively, are substantially different. In the same figures, the challenge of immunized mice and Salmonella pHCM1 and mice challenged with *Salmonella* pHCM1 with and without ampicillin are virtually the same. While this is not the only measure of the effect of immunization, the inconsistencies in the resulting survival curves should be addressed by the authors more thoroughly as they can confound the effects found in other parameters, including total and specific IgG and IgA, and pro-inflammatory cytokines.Overall the results are inconsistent and provide only partial evidence of the effectiveness of the RSP protein as a vaccine target.

To address the concerns regarding the disparities in survival rates depicted in Figures 1A and 3A, it is important to refer to several factors that contribute to these variations. Firstly, it should be noted that the data depicted in these figures stem from distinct experimental sets conducted at different times employing different batches of mice. Despite the use of the same strain and supplier, individual animals and their batches can exhibit variability in susceptibility to infection due to inherent biological differences.

Unlike in vitro cell culture experiments, which can achieve high replicability due to the homogeneity of cell lines, in vivo animal studies often exhibit greater variability. This variability is influenced not only by genetic variations within animal populations, even if originating from the same supplier, but also by environmental factors within the animal facility. These factors include temperature variations, the concentration y of non-pathogenic microorganisms in the facility, which can modify the immune responses, or the density of animals in the environment, consequently affecting human traffic and generating potential disturbances.

When designing experiments with animals, it is desirable for the results to be consistent across different animal batches. If one bacterial strain exhibits higher mortality rates than another across multiple experimental series, this pattern should be reproducible despite the inherent variability in in vivo studies. It is more important to demonstrate consistency in trends than to focus on absolute figures when validating experimental results.

It is also important to clarify that when we refer to survival rates, it doesn’ t necessarily mean that the animals were found deceased. The animal procedures were approved by the Ethics Committee of Animal Experimentation of the Universitat de Barcelona, which include an animal monitoring protocol. Our protocol requires close daily monitoring of several health and behavioral parameters, each evaluated according to specific criteria. When an animal reaches a predetermined score threshold indicating severe distress or suffering, euthanasia is administered to alleviate further suffering. At this point, biological samples are collected for subsequent analysis.

The conjugative experiments use very long conjugation times, making it harder to assess if the resulting transconjugants are the direct result of conjugation or just the growth of transconjugants obtained at earlier points in time. While this could be assessed from the obtained results, it is not a direct or precise measure.

In the conjuga2on experiments we u2lized a reduced number of donor cells expressing the RSP protein and of recipient cells, as well as long conjuga2on 2mes, to reflect more accurately a situa2on that may occur naturally in the environment. Short conjuga2on 2mes are efficient in controlled laboratory condi2ons using high densi2es of donor and recipient cells, but these condi2ons are not commonly found in the environment. For the interference of the conjuga2ve transfer of the IncHI plasmid we used an *E. coli* strain displaying the nanobody binding RSP to simulate a process that could be also scaled-up in a natural environment (i.e., a probio2c strain in a livestock farm) and that could be cost effec2ve. See discussion sec2on, lanes 326-328.

While the potential outcomes of these experiments could be applied to any bacterial species carrying this type of plasmids, it is unclear why the authors use *Salmonella* strains to evaluate it. The introduction does a great job of explaining the importance of these plasmids but falls short in introducing their relevance in *Salmonella*.

The prevalence of IncHI plasmids in *Salmonella* was indicated in the introduc2on sec2on, lanes 65-67. Nevertheless, we understand the reviewer’s cri2cisms and have modified both these sentences in the introduc2on sec2on and also added comments in the results sec2on (lanes 118-128).

**Recommendations for the authors:**

**Reviewer #2 (Recommendations For The Authors):**
I understand working with mice can be challenging in terms of repeating experiments to further support the study's claims. For this reason, I think the authors need to discuss more thoroughly the following things:Can the authors comment on why the presence of Ampicillin leads to a lower upregulation of proinflammatory cytokines in the spleen despite harboring resistance against ampicillin?

At the intestinal level, physiological inflammatory responses play a crucial role in enabling the host to identify foreign and commensal bacterial antigens and initiate a highly regulated and "controlled" immune response (Fiocchi, 2008. Inflamm Bowel Dis. 2008, 14 Suppl 2:S77-8). The administration of antibiotics such as ampicillin, reduces the load of intestinal resident microbiota, thereby lowering the extent of intestinal immune activation. This decline in immune activation extends to systemic levels, potentially accounting for the reduced expression of proinflammatory cytokines observed in the spleen.

There are inconsistent results in the survival rates in Figures 1A and 3A, please discuss how this could alter the observed differences in total and specific IgG and IgA, and pro-inflammatory cytokines.

To address the reviewer concerns regarding the discrepancies in survival rates shown in Figures 1A and 3A, and how these differences might influence the observed variations in total and specific IgG and IgA, as well as pro-inflammatory cytokines, it is important to clarify the terminology used in our study. In our context, "survival" does not solely refer to mortality per se, but encompasses the endpoints defined by our animal welfare protocols, which are rigorously supervised by the Animal Experimentation Ethics Committee of the University of Barcelona. Our protocol mandates close daily monitoring of several health and behavioral parameters, each scored according to specific criteria. When an animal reaches a predefined score threshold indicating severe distress or suffering, euthanasia is conducted to prevent further distress, at which point we collect biological samples for analysis.

In contrast to in vitro cell culture experiments, which often achieve high replicability thanks to the homogeneity of cell lines, in vivo animal studies frequently display greater variability. This variability stems not only from genetic differences within animal populations, even if originating from the same supplier, but also from environmental factors within the animal facility. These factors encompass variations in temperature, the presence of non-pathogenic microorganisms in the facility (capable of altering immune responses) and the density of animals, which can impact human traffic and potentially lead to disturbances.

The experiments depicted in Figs. 1A and 3A were separated in time, and hence may be influenced by environmental factors within the animal facility. Nevertheless, in the comparative analysis performed between immunized and non-immunized animals, experiments were performed simultaneously and hence under similar environmental conditions in the animal facility. For several parameters (i.e., immunoglobulins and proinflammatory cytokines) statistically significant differences were observed.

Regarding the conjugation assays, it is not entirely clear to me why the conjugation times are so long. It would be beneficial to have more data about the conjugation efficacy between the donor and recipient without any *E. coli* expressing the nanobodies at different time intervals. This would help to differentiate between transconjugants and transconjugants obtained from early conjugation events.

This comment is par2ally answered in a previous response, regarding the numbers of donor and recipient cells and dura2on of conjuga2on. We note here that in fig. 9, the requested experiment with donor and recipient cells without *E. coli* interferent cells is already present, corresponding to the label “none”. To avoid confusion, we have modified the legend in fig. 9.